# OpenAssistant Conversations - Democratizing Large Language Model Alignment

**Andreas Köpf**[*]
andreas.koepf@provisio.com

**Yannic Kilcher**[*]
yannic@ykilcher.com

**Dimitri von Rütte**      **Sotiris Anagnostidis**      **Zhi-Rui Tam**      **Keith Stevens**

**Abdullah Barhoum**      **Nguyen Minh Duc**      **Oliver Stanley**      **Richárd Nagyfi**      **Shahul ES**

**Sameer Suri**      **David Glushkov**      **Arnav Dantuluri**      **Andrew Maguire**

**Christoph Schuhmann**                    **Huu Nguyen**

**Alexander Mattick**
alexander.mattick@googlemail.com

## Abstract

Aligning large language models (LLMs) with human preferences has proven to drastically improve usability and has driven rapid adoption as demonstrated by ChatGPT. Alignment techniques such as supervised fine-tuning (*SFT*) and reinforcement learning from human feedback (*RLHF*) greatly reduce the required skill and domain knowledge to effectively harness the capabilities of LLMs, increasing their accessibility and utility across various domains. However, state-of-the-art alignment techniques like *RLHF* rely on high-quality human feedback data, which is expensive to create and often remains proprietary. In an effort to democratize research on large-scale alignment, we release OpenAssistant Conversations, a human-generated, human-annotated assistant-style conversation corpus consisting of 161,443 messages in 35 different languages, annotated with 461,292 quality ratings, resulting in over 10,000 complete and fully annotated conversation trees. The corpus is a product of a worldwide crowd-sourcing effort involving over 13,500 volunteers. Models trained on OpenAssistant Conversations show consistent improvements on standard benchmarks over respective base models. We release our code[2] and data[3] under a fully permissive licence.

A list of contributors who have chosen to be acknowledged by name can be found at https://open-assistant.io/contributors.

---

[*]These authors contributed equally to this work.

[2]https://github.com/LAION-AI/Open-Assistant

[3]https://huggingface.co/OpenAssistant/oasst1

37th Conference on Neural Information Processing Systems (NeurIPS 2023) Track on Datasets and Benchmarks.

# 1  Introduction

Artificial intelligence (AI), particularly in the field of natural language processing, has witnessed rapid progress in recent years. Major advancements are primarily driven by a straightforward formula: take a Transformer [1]-based architecture, increase the parameter count by enlarging depth and width, increase the size of the training corpus, and increase the scale of training compute. Although models have for some time exhibited an extraordinary ability to fit the training data and generalize based on their trained objective [2, 3], their adoption among the general public has until recently been slow. This can be mainly attributed to misalignment between model predictions and final intended usage.

The alignment of AI systems to human values, intentions, and preferences is a vital and intricate challenge within the AI research domain. This refers to the process of ensuring that AI systems can not only successfully optimize the provided surrogate training objectives, but also that their predictions are in line with their intended purpose and adhere to ethical and safety standards provided by humans [4, 5]. One possible solution is assistant-style fine-tuning of language models that has recently emerged as a promising approach to making large language models more in line with human preferences by generating more desirable outputs based on explicitly collected human preference data [6, 7, 8, 9, 10, 11] and thus making them more useful.

A notable instance of such an assistant-style model is ChatGPT, which has gained unprecedented user growth due to remarkable capabilities demonstrated in a wide range of fields, but also ease-of-use for the end user [12]. Aligning the model's predictions is in this case accomplished by introducing human-generated examples of intended usage and using reinforcement learning from human feedback [13, 14]. In *RLHF*, the human acts as a teacher and provides feedback in the form of rewards or penalties. In more detail, Ouyang et al. [13] proposed a three stage procedure to align language models: First, collect human-generated demonstrations of desired behaviour and train a supervised fine-tuned (*SFT*) model. Second, train a reward model (RM) on human-annotated rankings for different model outputs. Third, use the RM as a reward function and fine-tune the *SFT* model to maximize the reward generated by its responses. This is achieved using the PPO algorithm [15].

It becomes apparent that the benefits of all the aforementioned stages are predominantly dependent on the quality of the data used [16]. Despite this, availability of large-scale human feedback datasets for the open research community remains scarce. Most openly accessible datasets are comprised of synthetic data of instructions automatically generated by querying language models [17, 18, 19, 20]. Unfortunately, these datasets are limited with respect to their complexity, creativity and quality, as they rely on a pre-specified list of possible instruction types. Other datasets, such as Vicuna [21], use human-generated instructions, but still rely on langauge models to produce the respective responses. Without sufficiently broad and high quality data, even models with substantial size and pre-training would be inadequate for building capable, helpful, and harmless AI assistants.

Research in this area has predominantly been confined to a select few research labs with access to the required resources to engage in large-scale training and data collection. This monopolization of access to quality data undermines the potential for inclusive and diverse research endeavours, particularly in relation to alignment challenges, which arguably constitute some of the most crucial research areas of our time. In an effort to democratize research on aligning large language models, we introduce and release the OpenAssistant Conversations dataset. This dataset is the culmination of an extensive open- and crowd-sourcing initiative, and its release to the research community seeks to promote more inclusive research in this highly-influential domain. We provide a comprehensive analysis of the dataset, assessing ethical implications and safety considerations. We also fine-tune and release several assistant and preference models to further advance open access and research in this area. This transparency allows for iterative improvements on the released artifacts, fostering a more collaborative and inclusive research environment. By providing such a large and diverse dataset, OpenAssistant Conversations opens up new avenues of research in the field, enabling researchers to explore the complexities of human language and interactions in ways that were not possible before [22]. In the following sections, we delve into the intricacies of the OpenAssistant Conversations dataset and discuss its implications for the alignment of large language models and for society at large.

## 2 Data Format

The proposed dataset consists of a list of conversations. The basic data structure is a *Conversation Tree (CT)*, with nodes representing written messages in a conversation. A CT's root node represents an initial prompt, given by the prompter. To avoid confusion, we call the roles of the conversation *prompter* and *assistant*. This allows us to reserve the term *user* for the human contributors. Although our efforts focus largely on human contributions, both the prompter and assistant roles can, in principle, be fulfilled by either a human user or a machine. Every tree node is labelled by its role, and can have multiple children of the opposite role, each of which represents a separate next step in the conversation. A path from the root to any node in the tree (including to itself) is called a *thread*, and it represents a valid conversation with the prompter and the assistant taking turns. Tree nodes are annotated with additional data such as user-provided labels and metadata, such as collection timestamp and indicated language. Each *assistant* node further has a rank associated which orders it compared to replies of the parent prompt, according to user preferences.

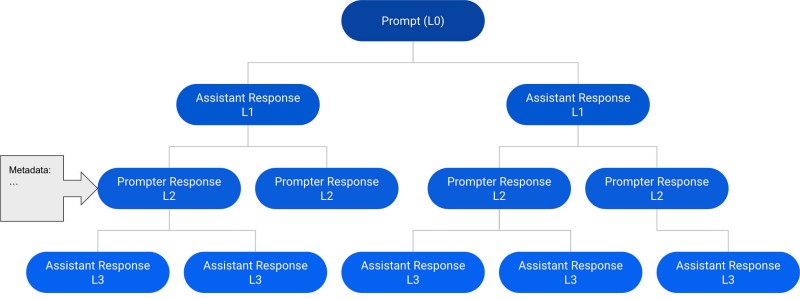

Figure 1: An example CT of depth 4 containing 12 messages. Any path from the root prompt to a node is a valid thread.

## 3 Data Collection

The OpenAssistant Conversations dataset is a comprehensive collection of conversational data that was obtained through a crowd-sourcing effort involving more than 13,000 volunteers. The data was collected using a web-app interface, dividing the collection of each tree into five separate steps: *prompting*, *labelling prompts*, *adding reply messages as prompter or assistant*, *labelling replies*, and *ranking assistant replies*. Users were fully informed about contributing to a public dataset. The dataset was curated with content moderation and spam filtering as key components of the annotation pipeline, ensuring high quality and safety standards.

Volunteers completed over 625,000 tasks in total, resulting in the collection of over 10,000 fully annotated and filtered Conversation Trees. Example User Interface (UI) displays of the data collection platform can be found in Appendix C, and current collection parameter settings, such as the number of collected replies to any parent message, can be found in Appendix G. In the following sections, we provide more details regarding the various aspects of the data collection pipeline.

### 3.1 Single-Step Collection

Data collection is structured to be both efficient and effective by breaking the work down into single units and advancing multiple conversation trees one step at a time. This approach minimizes data loss due to user attrition and ensures that every unit of work is captured for utilization. The users are presented with a range of task types, either by choice or through random sampling (weighted according to current requirements). The task types include creating prompts, replying as an assistant, replying as a prompter, labeling prompts or replies, and ranking prompter or assistant replies.

**Create a prompt.** Users write an initial prompt that forms the root of a new conversation tree. As this task is particularly popular, a lottery system manages the selection of new prompts, with only a fixed number of prompts being chosen for continuation at any given moment. This method serves to regulate the influx of new prompts and maintain a balanced distribution of tasks.

**Reply as assistant.** Replying as an assistant is a more labor-intensive task that necessitates users to carefully consider their responses and often engage in external research to provide a helpful and

relevant answer to the prompter's request. This task type, despite its demanding nature, has been reported to be the most enjoyable by many users due to the diverse array of topics covered. To account for the increased effort required for this task, a reward system has been implemented to incentivize users to participate. See Figure 6 for a UI preview.

**Reply as prompter.** The task of replying as a prompter, on the other hand, does not impose strict quality requirements but instead emphasizes on the importance of diversity to accommodate various use-cases. Examples of prompter replies may include asking for clarification, modifying the original intent, posing a follow-up question, or changing the direction of the conversation altogether.

**Label a prompt or reply.** Users are presented with a message from the database along with the preceding conversation thread (if available) and are asked to categorize the message according to three dimensions: spam detection, guideline adherence, and quality. For spam detection, users assess whether the message is unsuitable for inclusion in the dataset, for instances of obvious spam or trolling. Messages flagged as spam by multiple users are automatically removed from the dataset.

Guideline adherence is evaluated through a set of labels that determines whether the contribution aligns with the established guidelines (see Figure 4). These labels encompass the message being in a language other than the specified one, containing personally identifiable information, hate speech, sexual content, or being deemed inappropriate. Messages labelled in this manner are subsequently reviewed by human moderators.

Quality labels require users to rate the message on a five-point *Likert* scale across dimensions such as quality, creativity, humorousness, politeness, and harmlessness. These labels are stored for later analysis and application. Notably, users can voluntarily assign these labels (as well as spam & guideline adherence labels) to any message within the system, even as part of another task, as an additional contribution.

**Rank assistant replies.** Users are presented with two or more responses to the same parent message and asked to rank them in order of preference. This allows for a comparative analysis of the various responses and helps in identifying the most effective and engaging replies (Figure 5).

In summary, this data collection methodology effectively divides work into single units, minimizes data loss due to user attrition, and captures valuable information for future analysis and application. By offering users a diverse range of task types, the study encourages active participation and ensures the collection of rich and varied data for a comprehensive understanding of the subject.

## 3.2 Message Tree State Machine

The tree state machine serves as a systematic approach to managing the progression of message trees throughout the data collection process. This method ensures that each tree undergoes a series of states until it reaches completion, beginning with the creation of new trees by randomly sampling from the pool of initial prompts. The various states that a message tree passes through include the *initial prompt review state*, *growing state*, and *end state*, as well as the *aborted low-grade state* for trees that are deemed unsuitable for inclusion in the dataset and the *halted by moderator* state for trees that have manually been halted by a community moderator.

Upon the creation of a new tree, it enters the *initial prompt review state*, where multiple users are tasked with providing labels to assess its quality and suitability. This state plays a crucial role in identifying any potential issues with the initial prompt, and demands special attention, as the entire rest of the tree (potentially several dozens of tasks) is rooted in the initial prompt. If, at this point, the provided labels indicate that the tree contains spam or unsuitable content, it is transitioned to the *aborted low-grade state* and subsequently removed from the dataset. Conversely, if the tree passes the *initial prompt review state*, it proceeds to the *growing state*.

The *growing state* involves the continuous issuance of tasks to users, such as providing replies, labels, and rankings, to facilitate the development and expansion of the conversation tree. This state is essential for collecting diverse and rich data, as it allows for the accumulation of multiple interactions and the exploration of various conversation paths, given the same initial prompt. The *growing state* continues until the tree reaches its *end state*, which is defined by a maximum number of messages or other predetermined criteria.

Parameters within the data collection platform govern the behaviour of the tree state machine, such as the average number of messages collected for each parent message or the maximum tree depth. These

parameters enable researchers to fine-tune the data collection process according to their specific research goals and requirements, ensuring a more targeted and efficient approach to gathering data. Parameters varied during the collection of the dataset. Current settings can be found in Appendix G.

In summary, the tree state machine is a structured and systematic method for managing the progression of message trees during the data collection process. By guiding each tree through a series of states, from initial prompt review to growing and reaching its *end state*, the tree state machine ensures the collection of high-quality, diverse, and relevant data. Additionally, the inclusion of platform parameters allows for the customization of the data collection process to align with specific research objectives, further enhancing the effectiveness and utility of this approach.

### 3.3 Contributor Guidelines

To achieve a high degree of quality and consistency across a wide range of contributors, we issue clear and detailed guidelines. A full copy of these guidelines at the present time can be found in Appendix A. Our guidelines follow three main goals: First, clarify the meanings, scales, and criteria for assigning labels and rankings during the labelling and ranking tasks. Second, make assistant responses be polite, helpful, concise, friendly, and safety-aware and third, instruct prompts and prompter replies to explore a diverse and challenging set of inputs to the assistant role.

The guidelines establish a framework for safely interacting with an automated assistant by drawing inspiration from the concept of *informed consent*. Rather than categorically denying large parts of request categories, we aim to provide the prompter with useful feedback, for example drawing special awareness to dangerous activities, elaborating on weaknesses of automated assistants, such as hallucinations, and discouraging and denying requests asking for illegal or highly inappropriate content. In our validation experiments in training assistant models based on OpenAssistant Conversations, we observe a high degree of consistency of the trained models' outputs with our given guidelines.

Although guideline adherence is already high in our models after training, our approach is completely compatible with deploying additional safety measures during inference, such as secondary models to filter or modify ill-suited user input.

### 3.4 Quality Control & Content Moderation

We take a multi-pronged approach to quality assurance, with the main pillars being a system of reward points & leaderboards, and manual review of flagged content by human moderators. This both maximizes the quality of contributions, while effectively utilizing the limited time of the volunteer moderators. In an effort to demonstrate progress and achievement to users, and to encourage high-quality contributions, our system allocates points for the completion of tasks. These points contribute to various leaderboards, including daily, weekly, monthly, and all-time rankings. A level system also exists, wherein higher point accumulation results in elevated levels, reflecting veteran status and engagement. In the future, this system could potentially be developed further to facilitate preferential access to more engaging tasks or similar perks.

The distribution of points is contingent upon task type, as certain tasks require greater effort, such as the *reply as assistant* task (compared to the *create a prompt* task). A significant portion of points is deferred and reliant on interactions with other users. For instance, a user's assistant reply may gather many additional points if it is subsequently deemed non-spam and highly ranked by other users. Inversely, points may be reduced or lost for answers that are labeled as spam or down-voted by consensus of other users.

Within the moderator section of the website, an alternative leaderboard, designated the *Trollboard*, is exhibited. This leaderboard assesses users based on an aggregate of negative labels, reports, and down-votes received for their contributions. This approach enables human moderators to proactively scrutinize potentially misbehaving users in a comprehensive manner. The Trollboard has proven to be an effective tool in addressing the numerical disparity between users and moderators, maximizing the collective efforts of contributors to identify undesirable contributions.

Users further have the option to report messages to moderators for manual review, either via the platform, or directly via communication on a community chat server. Moderators have the ability to delete individual messages, or all messages of a given user, at their own discretion. Deleted messages are retained, but marked as deleted and are not exported for training.

## 4 Dataset Composition

The full dataset consists of 161,443 messages (91,829 prompter and 69,614 assistant messages) distributed across 66,497 conversation trees, in 35 different languages, annotated with 461,292 quality ratings. This includes 8,576 synthetic messages, leaving 152,867 human-submitted messages. Of the 66,497 total conversation trees, we consider 10,968 complete, meaning the full number of messages has been collected and the moderation process for these trees has been concluded. 52,159 incomplete trees are in the prompt lottery state, meaning they only consist of a single initial prompt. The completed trees contain 92,365 messages.

The set of categories for which *Likert*-scale human labels are collected is Creativity, Quality, Humor, Helpfulness, Violence, and Rudeness. The set of categories for which binary human labels are collected is Language Mismatch, Not Appropriate, Personally Identifiable Information, Hate Speech, and Sexual Content. We additionally release the rank of each assistant message compared to other assistant messages submitted for the same prompt, computed from the preference rankings of several human annotators. To merge the rankings of multiple (possibly conflicting) annotators, we use a variant of Tideman's method, described in Appendix B.

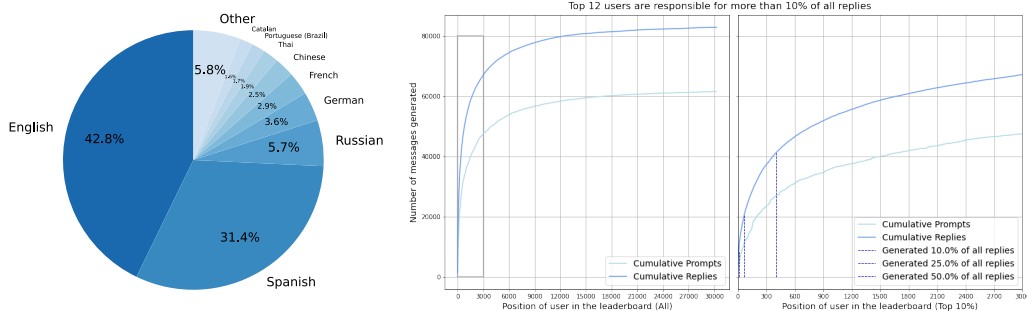

Figure 2: **Left:** Relative share of the most frequent languages in the dataset. **Right:** Distribution of contributions among users.

The dataset is dominated by English and Spanish messages as illustrated in Figure 2 (left). The prominence of English is expected as a result of the community around OpenAssistant originating in the English-speaking open-source machine learning community. The high quantity of Spanish messages can be attributed to the publicity given to OpenAssistant by prominent figures in the Spanish machine learning community. Figure 2 (right) illustrates how a small number of power users contributed a significant proportion of the dataset. This must be taken into account when considering possible biases in the data. Although significant effort went into preventing responses directly copied from other sources, it is possible that some users utilised automated techniques to enter data.

We release the dataset on the Hugging Face Hub[4] in several variants: One variant containing all collected initial prompts, one variant containing all trees that are considered completed, one variant containing all trees (completed and in-progress), and one variant containing messages filtered out as spam. For most purposes, such as instruction-tuning of language models, we recommend using the variant containing all completed trees, and include non-completed trees if more data is required.

## 5 Contributor Demographics and Satisfaction

To gain a deeper understanding of the contributors' demographics, a Google Form survey was sent out as an announcement on the project's Discord channel. The survey consists of 3 parts with questions on demographic information, personal motivation and user satisfaction. At the time of the release of this paper, a total of 270 survey answers have been collected. Results can be seen in Figures 7, 8, 9, 10, and Tables 3 and 4 (all in the Appendix). Respondent answers indicate a homogeneity towards identifying as male, but strong diversity in the reported levels of education, stated motivations for contribution, proficiency in artificial intelligence, and use cases for the technology. Over 95% of respondents either agreed or strongly agreed with the statement "Overall, I'm glad I have contributed

---

[4]https://huggingface.co/OpenAssistant/oasst1

to OpenAssistant." About 40% reported this being their first time contributing to an open-source project. We note that the method of recruiting via Discord is biased towards users who are present on the platform and have been active around the time of the announcement. Active contributors are also expected to be more likely to respond.

## 6 Experimental Validation

### 6.1 Instruction Tuning & Preference Modeling

We focus on the development and evaluation of fine-tuned language models based on Pythia [3], LLaMA [2], and Falcon [23]. Pythia and Falcon are state-of-the-art language models with permissive open-source licenses, while LLaMA is a powerful language model with a bespoke non-commercial license. Specifically we train supervised fine-tuned (SFT) models, reward models (RM [13],[24]), and, using the trained reward models, reinforcement-learned models (RLHF)[5].

| Model | LMEH | VEL | OAIE | HE |
|---|---|---|---|---|
| gpt-3.5-turbo (ChatGPT) | | 1110 | 0.87 | 0.72 |
| EleutherAI/pythia-12b | 60.33 | | | |
| OpenAssistant/pythia-12b-sft-v8-7k-steps | 60.28 | 997 | 0.10 | 0.10 |
| tiiuae/falcon-40b | 72.29 | | | |
| OpenAssistant/falcon-40b-sft-top1-560 | 74.04 | 1192 | 0.26 | 0.09 |
| OpenAssistant/falcon-40b-sft-mix-1226 | 74.40 | 1053 | 0.44 | 0.13 |
| huggyllama/llama-65b | 67.24 | | | |
| OpenAssistant/oasst-sft-7e3-llama-30b | 68.03 | 979 | 0.52 | 0.20 |
| OpenAssistant/oasst-rlhf-3-llama-30b-5k-steps | 68.51 | 1068 | 0.51 | 0.15 |

Table 1: Comparison of model evaluation scores on different LLM benchmarks: **LMEH:** lm-evaluation-harness [25] (average scores, see online leaderboard for more details) **VEL:** Vicuna Elo Rank [21] **OAIE:** OpenAI Evals [26] **HE:** HumanEval [27] (for all benchmarks, higher is better). We have chosen to leave the Hugging Face Hub identifiers as the model names for identifiability.

Table 1 shows evaluation scores of a selection of baseline and trained models on a set of standard benchmark datasets. Evaluations were performed externally using FastEval and evaluation results are hosted on a leaderboard, which is continually being updated[6]. LMEH refers to the average performance on a set of widely-used NLU tasks consisting of BoolQ, PIQA, HellaSwag, WinoGrande, ARC-e, ARC-c and OBQA[7]. We omit instruction-centric experiments (VEL, OAIE, HE) for the base models, as these benchmarks are unsuitable for non-instruction-tuned models. The results show that models using OpenAssistant Conversations are consistently outperforming the corresponding baseline models (in the case of LLaMA even a larger baseline model). RLHF outperforms SFT in some benchmarks, but not in others. For Falcon-based models, *sft-top1* is trained only on top-ranked conversation threads, whereas *sft-mix* mixes OpenAssistant Conversations with other instruction datasets (details in Appendix H). The varied evaluation scores demonstrate that by combining different data sources, the nature of the resulting model can be readily influenced. Ranks across benchmarks are not consistent, which could indicate the unsuitability of automatic evaluations for language models, or could indicate that different models and datasets lead to different capabilities. The results also show that while open-source models are close to matching ChatGPT in some benchmarks, others still show large performance gaps. Anecdotally, users report OpenAssistant models to be less robotic and more human-sounding than commercial models and report generations to have high quality and diversity in domains such as creative writing, conversational messaging, and drafting social media posts.

---

[5]All trained models are released at https://huggingface.co/OpenAssistant
[6]https://github.com/FastEval/FastEval, https://tju01.github.io/ilm-eval/
[7]For readability, Table 1 contains aggregated LMEH scores. Details in the online leaderboard.

## 6.2 Spam and Toxicity

To understand the concordance between human and automated toxicity detection, we employ toxicity detection methods based on Detoxify [28] to obtain automated ratings for six distinct categories, classifying whether a message is toxic, obscene, threatening, insulting, attacking a certain identity or sexually explicit. We limit our analysis to those languages that are supported by the toxicity detection method, covering English, Spanish, Russian, French, and Italian. These languages represent the majority of OASST1 messages (over 83%).

Using automated toxicity ratings, we are able to systematically assess the correlation between these ratings and human-assigned toxicity labels (hate speech, not appropriate, and sexual content). Based on a sample of 115,153 messages, we compute the correlation between automatic and human-annotated toxicity labels, which is visualized in Figure 3. We see a correlation between human and automatic labels in at least one element of each row and column of the correlation matrix, suggesting agreement between human annotators and off-the-shelf toxicity detection models. The results serve to both validate the capabilities and show limitations of AI-driven toxicity detection in comparison to human judgement and may inform future work in this area.



Figure 3: Correlation between human labels and Detoxify outputs for all messages in Detoxify-supported languages.

In addition to analysing the correlation between human-assigned toxicity labels and automated ratings, we extend the application of the Detoxify model to assess the efficacy of the moderation process for the same languages described earlier. To facilitate this analysis, we define two categories of messages: *deleted* messages, which encompass those that either failed to pass the community moderation process or were subsequently manually removed by moderators, and *retained* messages, which successfully made it through to the dataset. In order to provide a comprehensive evaluation of the moderation process, we calculated average values for each of the six Detoxify categories for both *deleted* and *retained* messages. The values obtained for this analysis are based on a sample of 74,781 messages. We excluded messages in trees that were incomplete at the time of export, as these messages may be subject to removal by the moderation process.

Our analysis, presented in Table 2 shows that the values for all six toxicity categories are markedly higher for *deleted* messages compared to *retained* messages. This significant difference demonstrates the effectiveness of the moderation processes in place, as messages removed from the dataset are on average rated as significantly more toxic by the Detoxify model than messages allowed to remain in the dataset.

| State | Toxicity | Obscene | Threat | Insult | Identity Attack | Explicit | N |
|---|---|---|---|---|---|---|---|
| Deleted | 4.625% | 1.965% | 0.411% | 2.085% | 0.651% | 1.39% | 3422 |
| Retained | 0.988% | 0.574% | 0.102% | 0.715% | 0.121% | 0.177% | 71359 |

Table 2: Detoxify outputs across six categories of toxicity, comparing *deleted* and *retained* messages.

While *deleted* messages are rated as more toxic than *retained* messages by the Detoxify model across all categories, the average toxicity values for these messages are still small. This implies toxicity ratings from models like Detoxify alone are not sufficient to determine when messages are unsuitable for inclusion in the dataset. Reasons for deleting non-toxic messages may include a lack of factual accuracy, or poor grammar. Additionally, messages which are children of deleted messages must themselves be deleted even if they appear to be acceptable in isolation.

# 7 Limitations

**Reward model data collection.** InstructGPT [13] trained reward models on ranking data of messages generated by their initial SFT model, while our reward models are trained using ranking data of human-generated messages. We chose to do so because we were already collecting this ranking data as part of our general efforts, for use in quality control, spam filtering, and dataset sub-sampling. While subjectively, many users report our RLHF models to follow instructions more closely (also compare Section 6.1), the models do not deliver the same uniform and significant improvements over SFT models as reported in [13]. We hypothesize that the difference in data collection for the reward model could at least partially explain this gap. We plan to collect ranking data based on our own SFT models in the future to verify these assumptions. Further research is necessary to determine more precise criteria for collecting data useful to RLHF.

**Subjective, Cultural, and Contribution Frequency Biases.** The open nature of our project introduces a unique set of challenges when it comes to controlling for biases within the dataset. Annotators from diverse backgrounds contribute to the dataset, with demographics that are simultaneously heterogeneous in some dimensions and homogeneous in others (see Section 5). Specifically, 89.1% of the annotators identify as male, with a median age of 26. This demographic profile may inadvertently introduce biases in the dataset, as it is bound to reflect the values, perspectives, and interests of the annotators. (We pose that some of this could be mitigated by introducing more constrained conversations, for example sampling a random Wikipedia page to determine the conversation topic.) Further, users' participation levels differ significantly. More engaged users contribute a greater number of annotations (see Figure 2), which likely leads to over-representation of their values and interests in the dataset. Consequently, the dataset may not adequately capture the diverse perspectives that a more balanced distribution of contributions could have provided. Further research is necessary to determine the effect of uneven contributor distributions have when given a clear, general task.

**Possibility of Unsafe Content.** While we have implemented measures to detect and remove harmful messages, our system is not infallible. It is possible that the dataset still contains unsafe content. We believe that the open nature of the project allows for data filtering to be conducted in a transparent manner, ultimately converging on the highest possible standards. Nevertheless, the potential presence of residual unsafe content in the dataset necessitates careful evaluation of any models trained on it.

Given the limitations discussed above, we advocate for the use of our models in academic research contexts only. We strongly encourage researchers to thoroughly investigate the safety and bias of any model before employing it in downstream tasks. The released models may exhibit unsafe behavior and are likely susceptible to prompt injection attacks. The alignment of LLMs is a crucial aspect of AI research, and we hope that our contributions can help advance the field of AI alignment. However, we also acknowledge that current alignment techniques are not perfect and can even exacerbate certain biases [29]. As such, researchers should exercise caution when using these models and be cognizant of their limitations. Additionally, it is essential to continue refining alignment techniques and advancing the field of AI alignment in order to mitigate these limitations and develop more reliable and robust LLMs.

# 8 Safety and Ethical Implications

Large language models are prone to generating inaccurate information about people, places, or facts, commonly known as 'hallucinations' [30, 31]. LLMs can also produce toxic or hateful content and fail to follow provided constraints [32]. Additionally, these models tend to incorporate biases present in their training data, leading to unfair and discriminatory outputs [33]. While methods such as *RLHF* can mitigate some of these shortcomings, they may exacerbate others [34, 29]. We hope that alignment methods using OpenAssistant Conversations can fix some of these issues [13], but we acknowledge that achieving full alignment is a complex and ongoing challenge.

We recognize that sufficiently powerful language models can have a significant impact on society [35], and therefore we believe it is essential to promote transparency in their development and deployment. OpenAssistant Conversations is our contribution to this goal of transparency.

## Acknowledgments and Disclosure of Funding

Our greatest thanks go to the many volunteer contributors, of human data, code, moderation, documentation, and community organization. Absent of any financial incentives, this project is a stunning and unprecedented display of global cooperation of humans for the purpose of advancing and democratizing AI research. In addition, several organizations have contributed to this project with resources: Redmond AI provided training compute. Stability AI and Hugging Face provided inference compute. We thank Olivier Dehaene at Hugging Face for close collaboration and personal support. Weights & Biases provided their full MLOps solution to the entire team. LAION provided legal input and acts as the website addressee. We thank Til Jasper Ullrich for running LLM benchmark evaluations, Luke Thomas Kaiser for running evaluations on model bias, and Silvia Pareti for detailed feedback on the manuscript. Agradecemos a Carlos Santana por la promoción a gran escala de la plataforma de recopilación de datos para la comunidad hispanohablante.[8]

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
