# OpenReview forum: "OpenAssistant Conversations - Democratizing Large Language Model Alignment"
_NeurIPS.cc/2023/Track/Datasets_and_Benchmarks — NeurIPS 2023 Datasets and Benchmarks Oral_

### Official Review · Reviewer_mgTk · 2023-07-16
**Great and important work, manuscript requires a bit of fine-tuning**

**Rating:** 9
**Confidence:** 4

**Strengths:**

The authors took special care to assure spam and fraudulent annotations were filtered out. Many datapoints and different types of labels were collected per document collected. The authors provide a detailed description of the data collection pipeline, enabling interested parties to extend their work. The data and the resulting models have been made freely available (with certain restrictions depending on their existing licenses), and the platform used creates confidence in the continued availability of the data. The data collection is still on-going, so the dataset size is expected to increase.

**Additional Feedback:**

I believe I’ve covered all my suggestions, thoughts and concerns in my previous responses.

**Clarity:**

While the paper is comprehensive for the most part, and details are abundant, there is some vagueness in some parts of the manuscript. For example, a lot of things are discussed in the beginning of the introduction (up to line 34), but very few references are included. Specifically, the authors include a definition of “alignment”. Presumably, they are not the first to define the term. Even if it is their definition (which should be explicitly stated, in that case), referring to previous work would be useful to readers. Moreover, they speak of the prevalence of ChatGPT. Aren’t there any resources documenting that? The authors are referring to something that is in the contemporary zeitgeist, which can alienate future readers. Finally (and I hate to be nit-picky), the authors begin by speaking of how the field of NLP within AI has “particularly” seen rapid growth. This is a quantitative statement that again is not backed up. I detect the same vagueness in some other places, such as lines 53, 106, or 275.

Since I agree with the authors on the importance of the task, I expect a lot of attention in their efforts from many disciplines, and so I believe less expert readers would appreciate more resources and concreteness. These can also future-proof the paper.

**Correctness:**

There is no absolute right or wrong way to construct such a dataset, so I think the design is sufficient. However, I highlight above deviations from previous work, which I think are worth discussing.

Moreover, it seems to me that the authors might be exaggerating the correlations discussed in Section 6.2. Is a correlation of .11 (column 3) really that large enough to claim anything? Or .2 in row 3? The readers are not given any context as to what constitutes a high or low correlation to make sense of the values. The toxicity values could be transformed into binary variables, or otherwise binned. What would the accuracy of the Detoxify be in that case, for example? Some sort of further analysis, like the aforementioned one or something else the authors deem more appropriate, to contextualize the results would really be helpful and serve to improve the paper and the validity of study, as well as inspire more trust in the lack of toxicity in the final dataset.

Finally, have the authors more deliberately studied the factual accuracy of the replies to get a sense of how well the volunteers could pick up to inconsistencies?

**Documentation:**

I see no issues with the documentation and the availability of the dataset in the future.

**Ethics:**

I see no major ethical implications in this specific work. The authors disclose limitations, which are generally applied to LLMs used as assistants, and not solely to their derived models.

**Limitations:**

The limitations are discussed, but I believe they need to be further highlighted, and perhaps some exploration of their effects rather than vague statements about values and perspectives would serve to improve the discussion. For example, are there any tangible, anticipated implications for the fact that most volunteers are male? Is there any relevant research on human personal assistants, like secretaries, that might elucidate the discussion? This is an interesting topic and some discussion of that in the paper could jumpstart research on the field if necessary.

**Opportunities For Improvement:**

The limitations w.r.t. the makeup of the people that participated in the data collection need to be emphasized further, and perhaps some of the figures from the supplementary material need to be transported into the main text. Were there any remediation strategies attempted to incentivize participation from other groups, and if so, are there insights as to why they failed?

Furthermore, it looks to me that the RM was trained on human responses to some prompt, rather than model generations. As far as I can tell, this is not the strategy followed by InstructGPT. Are there any particular reasons that the authors deviated from the strategy (apart from the fact that it requires the serialization of the data collection)? I believe this difference is worth explicitly highlighting in the paper. I also think that this is a possible reason the trained models are underperforming compared to ChatGPT, namely that there is some distance between what the RM has seen during training and what the SFT models produce. In effect, the training pipeline tries to successively approximate the desired behavior of the helpful assistant, and the ”domain” mismatch between the two models might be too large of a gap for the models to bridge successfully. Do the authors have any insights on this?

**Relation To Prior Work:**

The authors discuss how subsequent work from InstructGPT has not focused on data collection from people, and this work serves to remedy that. However, once again, I have highlighted in previous sections deviations from previous work, which I think are worth explicitly discussing.

**Summary And Contributions:**

The authors built a platform and a pipeline to allow collection of crowdsourced, high-quality data for the purpose of aligning LLMs to human preferences w.r.t. digital assistants. Their goal is to democratize both the usage and the research on such models given their contemporary prevalence.

---

> ### Author Response · Authors · 2023-08-21
>
> We thank the reviewer for acknowledging the rigorous effort to ensure the high quality of the data.
>
> > the makeup of the people that participated in the data collection
>
> We have experimented with moving the raw statistics into the main text, but that would need to be done holistically, and we haven't found a good way of doing so. However, our limitations section in the main paper goes extensively into this topic and highlights the important dimensions that contribute to imbalance (specifically age and gender, as other dimensions tend to be more spread out). We hope that this tradeoff is at least somewhat satisfactory.
> We have not undertaken any particular attempts to target specific groups, as we were focusing on getting contributions as such, and then on managing the influx of contributors.
>
> > the RM was trained on human responses to some prompt, rather than model generations
>
> Indeed the reward model for RLHF should in theory be run on rankings collected on outputs generated by an SFT–trained model, not on rankings generated by humans, as we did. We chose to collect these ranking data because it was a low-overhead way to collect ranking data that we felt was important for multiple reasons: It can be used as a quality signal in itself, it helps filtering the data if necessary, it can be used as an additional consideration for content moderation, and - even if suboptimal - it can be used for training a reward model. So initially, it was mostly a matter of practicality: We did not know how much longer we would get this volume high quality contributions, so we tried to collect as much as we can, rather than waiting until we figure out good SFT training recipes. At that time, we did not yet know how large the difference would be for the resulting model performance between the two collection methodologies. Our experience shows that the difference is maybe larger than thought, and while our RLHF models exhibit certain desirable qualities, they are not a clear-cut overall winner. However, they can certainly serve as a lower bound on future RLHF efforts. We have amended the limitations section with a section discussing this. Of course, nothing of this is preventing us from collecting model-based ranking data in the future, which we plan to do.
>
> > expanding on the limitations with more precision
>
> We agree, many questions are raised and not answered, and perhaps not served to the best possible degree by e.g. just highlighting a difference in contributions. You bring up an excellent question: Is the question "how do I expect a helpful universal assistant to behave" answered significantly differently by males and non-males? And if yes, is that difference significantly larger than the variance among the individual groups? Such questions (and many more) would need to be considered to answer definitively whether any imbalance in our contributor makeup does actually constitute an effective limitation on our work. From our (limited) overview of the research landscape, we haven't found any conclusive answer to these questions, so we feel that opening this discussion in an academically satisfactory manner might be better left to derivative or complimentary work. We have amended the limitations section and hope that can give at least a small impetus for such work.
>
> > exaggerating the correlations discussed in Section 6.2
>
> The idea behind these values is slightly double-edged: We aim to show that while human moderation does largely go into the same direction as automated detection systems, such systems are as of yet too limited to cover everything that is necessary to match human judgements. The general direction of the numbers - especially in cells that semantically match - is aiming to show the first part, while the still small absolute values are aiming to show the second part, which we tried to capture in our discussion of these results.
>
> > have the authors more deliberately studied the factual accuracy of the replies to get a sense of how well the volunteers could pick up to inconsistencies?
>
> Quantitative tests of correctness are naturally hard to do due to the intractability of estimating factual accuracy in free-form text. Qualitatively, users seem to be very dedicated to factual correctness, often citing or quoting information from other resources. The random assignment of answering tasks could be a contributor to this as one frequently gets confronted with questions that, while not difficult in the absolute, needs a little bit of research, which people tend to cite to increase the likelihood of “winning” the answer-quality benchmark.
> To answer the question: We have not systematically studied the factual accuracy of the contributions, but punctually and via reports from other contributors, we have done so deliberately.

---

> > ### Author Response · Authors · 2023-08-21
> >
> > > Vagueness in the paper
> >
> > This is a very valid criticism. We will expand our citing efforts when it comes to ChatGPTs popularity and alignment: This is very much a situation where we assumed readers had sufficient context, but especially vague statements like “alignment” or “prevalence of ChatGPT” should be more well grounded to preserve context for readers from different domains or times.
> > Changes are not yet in this update version, but we will include them in the next update.

---

> > > ### Comment · Reviewer_mgTk · 2023-08-24
> > >
> > > Thank you for your response. My concerns were mostly addressed.

---

### Official Review · Reviewer_dMbi · 2023-07-20
**Promising potential meets challenges: a nice initiative requiring refined evaluation methods and experiment design**

**Rating:** 7
**Confidence:** 3

**Strengths:**

- Ambitious and democratizing initiative that's of interest to the NLP community.
- Large-scale dataset with 161K chit-chat style messages across 35 languages.
- Rigorous data quality measures, including spam filtering and content moderation on multiple dimensions.

**Additional Feedback:**

No

**Clarity:**

While the paper maintains a smooth flow, it falls short in providing essential explanations for certain sections.

**Correctness:**

The dataset evaluation lacks correctness. The work does not provide a clear explanation of the evaluation metrics and benchmarks that were used to assess the performance of the models accurately.

**Documentation:**

Yes

**Limitations:**

Yes, the authors have adequately addressed the limitations and potential negative societal impact of their work.

**Opportunities For Improvement:**

Weaknesses (in roughly decreasing order of importance):

- **Lack of evaluation metrics in Section 6.1**: The results in Table 1 lack clarity on how the models trained on collected data were evaluated. The mention of evaluation by "Til Jasper Ullrich" lacks explanation and context.
-  **Absence of benchmark explanation used for evaluating the trained models**: for example, I'm not sure what LMEH refers to.
- **Incomplete comprehensive evaluations** due to computational resource shortage: The authors acknowledge missing experiments due to limited computational resources at the time of writing, raising concerns about the paper's acceptance without crucial evaluations.
- **Absence of human evaluation results for response quality in OpenAssistant models**: While the authors mention user reports on positive aspects, such as improved naturalness and diversity compared to ChatGPT, there is no presentation of human evaluation results to substantiate these claims.
- **Unclear language context for reported results**: The paper does not specify whether the reported results were conducted on English or other languages, leaving room for ambiguity.
- **Insufficient related work section**: The introduction briefly mentions previous work, but lacks a comprehensive and rigorous analysis of related literature.
- **Ambiguity on open-ended conversations and instruction tuning**: The paper does not sufficiently clarify how open-ended conversations are relevant to instruction tuning.
- **Lack of clear criteria for evaluating response quality**: The paper does not provide a clear definition of what "quality" entails when assessing responses. It does not specify whether it refers to grammatical correctness or other aspects of response evaluation.
- **Absence of conversation examples in the main paper**: It would be beneficial to include an example of the conversation early on in the paper, and consider moving some of the data collection details from the appendix to the main paper for better accessibility.

**Relation To Prior Work:**

No, the work does not clearly discuss how it differs from previous contributions.

**Summary And Contributions:**

The paper introduces an ambitious initiative aimed at democratizing research on large-scale alignment by presenting the "OpenAssistant Conversations" benchmark. This benchmark consists of a collection of 161K chit-chat style messages, gathered through a worldwide crowd-sourcing effort involving over 13K workers across 35 languages. To ensure data quality, the authors implemented rigorous checks, filtering out spams, and employing content moderation on various dimensions, such as creativity, humorousness, politeness, harmlessness, language adherence, hate speech, and sexual content.
The conducted experiments demonstrate the effectiveness of the moderation process in generating high-quality messages. Additionally, the authors fine-tuned LLaMA and Falcon on the collected benchmark and observed significant performance improvements.

---

> ### Author Response · Authors · 2023-08-21
>
> We thank the reviewer for their extensive feedback. In the following, we address the concerns made.
>
> > Lack of evaluation metrics in Section 6.1
> > Absence of benchmark explanation used for evaluating the trained models.
>
> Evaluating the performance of LLMs and the degree of the achieved alignment based on values, intentions, and preferences set beforehand is in general challenging. In this work, we have included several benchmarks (BoolQ, PIQA, HellaSwag, WinoGrande, ARC-e, ARC-c, OBQA; all as part of LMEH) that are widely adopted by the community and are commonly used to evaluate LLM performance (see e.g. [1, 2]). For the purposes of this work, we believe that these benchmarks do not wholly capture the capabilities of conversational agents. For this reason, we have included the Vicuna Elo Rank, OpenAI Evals, and HumanEval benchmarks. We agree with the reviewer that a more detailed description of the tasks used will be beneficial for the reader. We will include such a discussion in the Appendix in the next days.
> We also updated the paper to clarify the meaning of LMEH: It refers to the unweighted average performance on the aforementioned NLU tasks, consisting of BoolQ, PIQA, HellaSwag, WinoGrande, ARC-e, ARC-c and OBQA, and stands for Langue Model Evaluation Harness.
> We have also clarified the origin and methodology of the evaluations.
>
> [1] Dettmers, Tim, et al. "Llm. int8 (): 8-bit matrix multiplication for transformers at scale." arXiv preprint arXiv:2208.07339 (2022).
>
> [2] Frantar, Elias, and Dan Alistarh. "Massive language models can be accurately pruned in one-shot." arXiv preprint arXiv:2301.00774 (2023).
>
> > Incomplete comprehensive evaluations
>
> We agree. We have now included the missing evaluation of the Pythia baseline model.
> Note that the remaining non-filled cells in the table refer to instruction-centric evaluations (VEL, OAIE and HE). Those have been omitted for the base models, as these benchmarks are unsuitable for non-instruction-tuned models and resulting numbers would not be meaningful. The LMEH numbers allow comparison of base models to instruction-tuned models, while the instruction-centric numbers allow for comparison among such instruction-tuned models. We agree that the wording in the paper is ambiguous and have updated the paper to clarify this.
>
> > Absence of human evaluation results for response quality in OpenAssistant models
>
> Indeed, our statements on aspects such as naturalness and topical diversity of the resulting models are anecdotal and subjective, which we point out in the paper. Assessing these qualities of the models trained on OASST1 would certainly be insightful, but it is somewhat out of scope for this work. Our focus is the dataset itself, not the models, and for the dataset we believe that our extensive collection of quality labels (thumbs up/down), labels of specific dimensions (helpful, humorous, etc.), and the additional collection of answer rankings provide a plethora of human-created data on the quality of the collected data, all of which is released as part of OASST1.
>
> > Unclear language context for reported results
>
> We followed standard procedures for all experiments and carried out the evaluations in the original language of each respective benchmark, which to the best of our knowledge is predominantly English. However, both the evaluation benchmarks and the models are inherently language-agnostic and don't require the selection of an explicit language to work in.
>
> > Insufficient related work section
>
> Thank you for providing this feedback. Upon suggestion of this and Tg3T's feedback, we have extended the related work section and will continue to do so in the next days.

---

> > ### Author Response · Authors · 2023-08-21
> >
> > > Ambiguity on open-ended conversations and instruction tuning
> >
> > There is indeed a degree of ambiguity here, which we did not sufficiently address. We use the term "instruction tuning" broadly to refer to the process of fine-tuning pre-trained language models with instruction-style data. While our collected data is open-ended from a topical perspective, it is not "conversational data" per se, as it is not simply a record of a conversation, but specifically a conversation between a prompter with some task, information request, or goal, and a helpful assistant that attempts to fulfill these tasks and requests, which falls into the category of instruction-style data.
> > So while we don't rigorously enforce that every input is an instruction, by construction the dataset is naturally instruction-style, rather than plain conversational.
> >
> > > Lack of clear criteria for evaluating response quality
> >
> > The assessment of response quality is determined in our contributor guidelines, which have a set of baseline criteria that users should follow when evaluating responses, these include, but not limited to:
> > Factual accuracy and helpfulness are first and foremost.
> > Don't Judge quality based on personal beliefs
> > Penalize replies that fail to provide adequate warnings or caveats
> > Penalize replies that are difficult to read due to a lack of formatting, capitalization or other errors.
> > Penalize replies if the requested information is obfuscated by superfluous details that make up a large part of the message
> > Don’t Rank replies based on how long and short they are - instead, find out which reply best answers the query of the user.
> >
> >
> >
> > > Absence of conversation examples in the main paper
> >
> > That's a good idea. We've experimented with moving things around, but have not found a satisfactory way to do so in the available 9 pages. Should this paper get accepted, we can use part of the additional page for this.

---

> > ### Comment · Reviewer_dMbi · 2023-08-27
> > **Thanks for the detailed explanation**
> >
> > I thank the authors for the thorough explanation. They have effectively addressed my concerns.  As such, I have raised my score to 7.  I believe the work is robust and deserves acceptance. The dataset will provide significant value to the NLP community.

---

### Official Review · Reviewer_EoWL · 2023-07-21
**Opensourced human-annotated conversation dataset collected for training conversational AI systems.**

**Rating:** 9
**Confidence:** 4
**Correctness:** Correct.
**Clarity:** Yes.

**Strengths:**

- Unique in its open crowd-sourced nature involving thousands of worldwide volunteers. Helps democratize research on aligning large language models by releasing data openly.
- Rigorous analysis of dataset statistics and contributor demographics. Experimental validation showing performance gains over base models.
- Attempts to assess and thoughtfully discuss limitations like biases in the data. Overall goal of transparency and democratization has positive implications.

**Additional Feedback:**

The key strengths are the significance of releasing this large human-annotated dataset openly to the research community, combined with strong efforts to validate the data's utility and discuss its limitations transparently.

**Documentation:**

Yes.

**Ethics:**

No.

**Limitations:**

Limitations are adequately discussed and studied.

**Opportunities For Improvement:**

- Conversational AI is a huge area - this data may not cover all subdomains. Authors are encouraged to add statements regarding this and explain how diversity among prompts and replies could be achieved.

- While a large dataset, even bigger with more diverse contributors could be more impactful. It is valuable to outline the planning for the next version.

- Additional modalities like image and video could make the conversations richer in the future :-)


**Relation To Prior Work:**

Yes.

**Summary And Contributions:**

- Releases a dataset of 161,443 human-generated dialog messages in 35 languages, annotated with over 460,000 quality ratings. This represents a significant scale of high-quality human feedback data.
- The data was collected through a worldwide volunteer effort involving over 13,500 contributors, making it a unique open crowd-sourced dataset with ensured diversity and transparancy.
- Shows experimental results training models with the data, demonstrating improved performance over base models on standard benchmarks. Models and code are also released.
- Aims to promote more open, inclusive research on aligning large language models.

---

> ### Author Response · Authors · 2023-08-21
>
> We thank the reviewer for the positive feedback and for sharing our enthusiasm for promoting open, inclusive research. We truly believe that only through the open and accessible collaboration of different research communities, LLMs will truly become more useful, while mitigating potential harms.
>
> > Diversity between prompts and replies
>
> We fully agree, diversity in subdomains is a major factor in providing helpful general assistant models. We have attempted to showcase the topical diversity to a degree in the word clouds in the Appendix, but haven't done a systematic investigation beyond that. As for mitigation, we have several ideas, such as sampling random Wikipedia pages to determine the conversation topic at the outset. We have amended the limitations section by a statement regarding this.
>
> > Plans for the future
>
> We are really excited about future versions of the datasets and models. We are planning to continue releasing updated versions of the dataset, as more samples are collected. Provided that the need arises, the format of our dataset, i.e. conversational trees, allows for the introduction of other modalities as well, including that of images and videos!
> We want to point out, however, that we did this during a very unique time in ML history, which presented us an opportunity to motivate this many people for contributions. As we all know, life moves on, and something new will take center stage in people's minds, such as tiny levitating rocks. We expect the volume of contributions to decrease, but we'll adjust and make the best of it. We feel that OASST1 (and its successors) is a contribution that even years from now can still serve as a vital ingredient in training conversation systems, and by that we hope to have extended this brief moment of worldwide excitement to a much longer timespan.

---

> > ### Comment · Reviewer_EoWL · 2023-08-29
> >
> > Many thanks for the authors to contribute to the community again. The plan and amendment regarding diveristy looks good to me.

---

### Official Review · Reviewer_GdLf · 2023-07-21

**Rating:** 9
**Confidence:** 4
**Correctness:** Yes.
**Clarity:** Yes.

**Strengths:**

1. The scale and diversity of the dataset is impressive, especially in terms of the number of annotated languages. This helps mitigate demographic biases compared to other existing datasets like Alpaca / Anthropic-HH / sharegpt.

2. Releasing the data and trained models openly is a significant contribution to democratizing research on aligning large language models.

3. The data collection methodology using single-task contributions and a state machine is well-designed.

4. The analysis of the correlation between human and automated toxicity ratings provides valuable insights. The results serve to validate the capabilities and show limitations of AI-driven toxicity detection and may inform future work in this area.

**Additional Feedback:**

Questions:

1. what is the design principle behind Conversation Trees in OpenAssistant compared to thread-like data in sharegpt? Since many models are only trained on the single-round subset of OpenAssistant data [1], this effectively reduces the number of data that can be used to train a single-round AI assistant.

[1] QLoRA: Efficient Finetuning of Quantized LLMs

Tiny problesm:

line 45: Most openly accessible datasets are comprised of synthetic data of instructions automatically generated by querying language models [9, 10, 11, 12, 13]. ==> Vicuna should not be cited here since its instructions are provided by users to ShareGPT.

**Documentation:**

Yes.

**Ethics:**

No.

**Limitations:**

Yes.

**Opportunities For Improvement:**

1. The reinforcement learning from human feedback (RLHF) experimental validation is limited in utility, as the reward model used for RLHF is not trained on preference rankings collected from the same base model that is then fine-tuned. Using rankings from a different model for the reward signal likely diminishes the effectiveness of RLHF.


**Relation To Prior Work:**

Yes.

**Summary And Contributions:**

This paper introduces OpenAssistant Conversations, a large-scale human-generated dataset for training open-domain dialog agents. The dataset contains over 150,000 human-written conversational exchanges in 35 languages, annotated with quality ratings and other metadata. The authors demonstrate the utility of the dataset by training several open-source language models which achieve improved performance compared to their base versions.

---

> ### Author Response · Authors · 2023-08-21
>
> We thank the reviewer for the positive feedback and for sharing our enthusiasm about this project. Collecting samples in a plethora of different languages was a direct conclusion of the amazing community that developed around this project. We are particularly excited to expand this set to ever great lengths.
>
> > The reinforcement learning from human feedback (RLHF) experimental validation is limited in utility, as the reward model used for RLHF is not trained on preference rankings collected from the same base model that is then fine-tuned. Using rankings from a different model for the reward signal likely diminishes the effectiveness of RLHF.
>
> This is definitely true. Our efforts at RLHF are still in very early stages as our focus was primarily on creating a good SFT dataset and SFT model in the first place, to then iterate using e.g. RLHF based methods. One should see the RLHF results as a lower-bound to what could be possible in future versions of OpenAssistant. We have amended the limitations section with a paragraph discussing this further.
>
> > What is the design principle behind Conversation Trees in OpenAssistant compared to thread-like data in sharegpt?
>
> Conversation trees are the natural data structure if one considers different users working on independent threads. This is a stricter requirement than the one in sharegpt, where one can assume a single contiguous conversation held between a user and ChatGPT. If we constrained OpenAssistant to a similar “single contiguous conversation” standard, the conversation tree datastructure would have a branching factor of 1 and become a thread-like object just like sharegpt. However, since our objective is getting multiple human answers to the same questions to increase the diversity of opinion and background, a tree like datastructure with dependent conversation flows is the natural choice for our dataset.
> In general, we have not observed that training on multiple threads from the same tree is bad for the model (as in, the beginning of the threads would be repeated in different samples), rather the single thread subsets often limit themselves to always take the top-ranked thread of any conversation tree, and therefore drastically increase the average quality of the data. Even if that results in the data size being lower than the original data, the increase in quality more than makes up for it. In light of this, collecting multiple answers at each node, combined with the ranking data, is still not superfluous, rather it is the enabling factor to later filter the data for high quality samples. (Although we concede, if that was the only goal, one could definitely go about it in a smarter way)
>
> > Vicuna dataset
>
> Thank you for pointing this out. We have updated the text to highlight the differences in the Vicuna dataset.

---

> > ### Comment · Reviewer_GdLf · 2023-08-30
> >
> > I thank the authors' response. My impression of this paper remains quite postive

---

### Official Review · Reviewer_Tg3T · 2023-07-23
**A very important contribution to the field**

**Rating:** 9
**Confidence:** 5
**Correctness:** The paper is sound and highly transpa…
**Clarity:** The presentation is excellent

**Strengths:**

OpenAssistant Conversations being one of the largest open human feedback datasets is clearly a very important contribution to the field and the open LLM research. The dataset opens new avenues for experiments in alignment with human values and for LLM improvement in general. I would like to thank the authors for this incredible work, I very much enjoyed reading this paper.

Strengths:
1. The authors introduce and release a dataset of conversations with more than 160000 annotated messages, with over 10000 fully annotated conversation trees in 35 languages
2. The dataset construction is discussed in great detail, with excellent explanations of all design choices, of the system and UI for annotations, and of the instructions provided for the annotators. This is an important contribution in itself as it allows other researchers to replicate and improve the presented approach for their own dataset collection.
3. The authors also present experiments with models trained on the dataset and release 20 pretrained models including one RLHF-ed model and reward models. All the code is released and both the training code as well as the provided checkpoints are very valuable for the community.

Without a doubt, this paper is a clear accept.

**Additional Feedback:**

See Questions in Opportunities For Improvement.

**Documentation:**

The level of detail on dataset construction and documentation is excellent.

**Ethics:**

I do not see any ethical concerns, the paper transparently addresses limitations and societal impact

**Limitations:**

Limitations and societal impact are addressed.

**Opportunities For Improvement:**

While the paper is excellent, a few things can be improved. I also have a few questions.

Weaknesses:
1. Related work can be improved as some other human preference datasets already exist. For example, Anthropic's RLHF dataset [1] as well  as Stanford Human Preference Dataset [2] should be mentioned.
2. Experimental results with models do not include the Pythia baseline model, therefore it is hard to see the improvement provided by fine-tuning the model on OpenAssistant Conversations
3. While OpenAssistant Conversations is a dataset with human feedback annotations, only a single RLHF-ed model (LLAMA-based) is released and included in experiments. The release of other RLHF-ed models and their inclusion in experiments would be a huge benefit to the community, especially given the research-only license of the first version of LLAMA.
4. Additionally, even for the LLAMA-based RLHF-ed model the benefits of RLHF are somewhat unclear because of the inconsistency across benchmarks. While the authors say that "this could indicate the unsuitability of automatic evaluations for language models, or could indicate that different models and datasets lead to different capabilities", it could be a reflection of biases in different benchmarks. It would be interesting to see a more detailed discussion of the RLHF results compared to the SFT model.
5. Details on "sft-mix" model are missing other than "it mixes OpenAssistant Conversations with other instruction datasets". It is important to provide these details in the paper and cite those other datasets.
6. Reward model architecture details and performance evaluation is not presented in the paper. It is important to provide details on reward model architecture and design choices and present their test performance in ranking assistant responses according to human preferences.

Questions:
1. Could you please comment on stability of RLHF training? How robust were RLHF results?
2. One of important possible limitations of the dataset could be the undetected use of ChatGPT by annotators. What was your quality control approach to making sure that users did not use ChatGPT?
3. Why were the assistant replies in the dataset produced by humans? Why were LLMs (for example, the LLAMA, Falcon and Pythia baseline models) not used for that purpose?
4. Could you please expand your point: "The varied evaluation scores demonstrate that by combining different data sources, the nature of the resulting model can be readily influenced."? What kind of changes of model nature did you observe in your results? What would be the recipe for an intentional change of the nature of the model?
5. In the section on toxicity detection, you say: "The results serve to validate the capabilities and show limitations of AI-driven toxicity detection and may inform future work in this area". While the promise of automated toxicity detection is clear from this section, what are the main limitations and opportunities for improvement in your view?
6. In the limitations you mention:"We believe that the open nature of the project allows for data filtering to be conducted in a transparent manner, ultimately converging on the highest possible standards". While this is an excellent point, is there a process in place to allow external users and open source community to take part in data filtering so that it does indeed converge to the highest standard?


References:
[1] Paper: Bai, Y., Jones, A., Ndousse, K., Askell, A., Chen, A., DasSarma, N., Drain, D., Fort, S., Ganguli, D., Henighan, T. and Joseph, N., 2022. Training a helpful and harmless assistant with reinforcement learning from human feedback. arXiv preprint arXiv:2204.05862.
Dataset: https://huggingface.co/datasets/Anthropic/hh-rlhf

[2] Stanford Human Preference Dataset: https://huggingface.co/datasets/stanfordnlp/SHP

**Relation To Prior Work:**

Relation to prior work is covered, however, can be improved.

**Summary And Contributions:**

The paper presents OpenAssistant Conversations -- a large dataset of dialogue conversations with human feedback annotations. The dataset is released under open source license and greatly improves transparency of and access to high quality large-scale LLM research resources. The dataset includes more than 160000 annotated messages with over 10000 fully annotated conversation trees (in 35 languages). OpenAssistant Conversations dataset enables RLHF and novel alignment experiments. The paper details the construction of the dataset including the interface and instructions used for annotation, as well as presents example models trained on the dataset showcasing its promise. This is a unique global collaborative effort of over 13500 volunteers. The dataset and code are released both on Hugging Face and Github.

---

> ### Author Response · Authors · 2023-08-21
>
> We thank the reviewer for the valuable feedback and suggestions to improve our work. We are really excited to share this collaborative effort with the open-source community and are happy to see other people equally excited about it. In the following we address the concerns raised.
>
> > Related work can be improved
>
> Thank you for bringing this up. We have added the suggested citations. We will also be looking for more.
>
> > Experimental results with models do not include the Pythia baseline model
>
> We agree that presenting results for the base pythia model would serve as a baseline to understand differences between finetuned models. For completeness, we present results for the base pythia model, complementary to Table 1 and updated the paper.
>
>
>
> |Task (LMEH)|Performance|
> |-----------|-----------|
> |BoolQ      | 65.87 |
> |PIQA       | 77.04 |
> |HellaSwag  | 68.83 |
> |WinoGrande | 65.59 |
> |ARC-e      | 66.62 |
> |ARC-c      | 38.14 |
> |OBQA       | 40.20 |
> |Average    | 60.33 (<-- this is the LMEH number for the paper) |
>
> As expected for LMEH tasks, performance between base and finetuned models is very similar. Note that the other benchmarks are specific to instruction-tuned models and are not evaluated for base language models.
>
> > Release of more RLHF-ed models
>
> As we discuss below, RLHF on our data showed not as much benefit as we expected, thus we did not produce very many RLHF models.
> But we agree, more should exist, so we have now also made our Pythia-based RLHF model public on the Hugging Face Hub for reference. However, we also encourage people to use our training code and create their own.
>
> > The benefits of RLHF are somewhat unclear
>
> Indeed, that is an observation we made as well. Anecdotally, RLHF models tend to be favored by humans for certain tasks, while minimally affecting the model's general knowledge. But we did not find a consistent and significant improvement in the metrics for RLHF models. Further, we found RLHF training to be more brittle than SFT. So, given unclear improvements, combined with unstable training, we largely focused on finding better dataset mixes for SFT runs.
>
> The natural question is: Why does RLHF seem to help in the original InstructGPT paper and less so in our case? We suspect, at least partially (and as the reviewers here have also noted), while the collection of our conversation data follows the collection of InstructGPT's conversation data, the collection of our RLHF data diverges slightly from InstructGPT's: They collect rankings on SFT model outputs, while we collect rankings on the human conversation data. (We did this because collecting conversations and rankings allowed us to make optimal use of the influx of volunteers, and gather extra quality labels for the conversation data.) This difference in collection methodology could explain some of the different effects of RLHF. We plan to utilize the feedback we gather from human-model interactions to investigate the direction of RLHF more in the future. We have amended the limitations section by a paragraph discussing this.
>
> > Details on "sft-mix" model are missing
>
> Reproducibility was and continues to be a high priority. We are continuously researching the effects of dataset composition, of which sft-mix is one instance. We have added its exact composition in the appendix.
>
> > Reward model details
>
> Details regarding training and models are presented in Appendix G. The reward is derived by adjusting a linear layer on the predictions of a base model, that generates a scalar prediction (see also L877-884 in Appendix G). Performance is measured by measuring the ability to predict the better reply among pairs of replies with different rank, on a held-out validation set. We have amended this in the appendix.
>
> > Could you please comment on stability of RLHF training? How robust were RLHF results?
>
> We found that RLHF quickly overfitted to the reward model that was used to train with PPO. Different regularization methods are taken into consideration, which are outlined in the open-source code. We are also planning on releasing more w&b runs publicly to offer more insights. Further, we have additionally made a Pythia-based RLHF model public on the Hugging Face hub for reference.

---

> > ### Author Response · Authors · 2023-08-21
> >
> > > One of important possible limitations of the dataset could be the undetected use of ChatGPT by annotators. What was your quality control approach to making sure that users did not use ChatGPT?
> >
> > This is indeed one of our major concerns. Our guidelines https://projects.laion.ai/Open-Assistant/docs/guides/guidelines#dont specifically mention this point as the first item in the list, discouraging contributors from copy-pasting responses generated by other AI models.
> > Users who were found to post ChatGPT-generated responses were banned, and their contributions were deleted. Furthermore, we used multiple automatic tests to catch such cases. For instance, we searched for and removed messages that contained text such as “as a large language model” or “knowledge cutoff after September 2021”.
> > Moreover, users were encouraged to up-vote and down-vote responses they came across from other users, which also helped weeding out low-quality, generic, (possibly AI generated) responses.
> > We have added a paragraph discussing this to the appendix.
> >
> > > Why were the assistant replies in the dataset produced by humans?
> >
> > The main objective of our first iteration of open-assistant is to generate high quality reference data to be used in supervised finetuning. In this initial stage the focus is to align a model’s output to what a human would consider a “gold standard”, which is why we focus on human created replies.
> > We plan on future iterations utilizing answers from pre-existing models, which we expect to produce better signals for future reinforcement learning datasets, as the reinforcement learning stage is more focused on specifically penalizing model-induced errors, rather than giving a ground-truth for alignment.
> >
> >
> > > Could you please expand your point: "The varied evaluation scores demonstrate that by combining different data sources, the nature of the resulting model can be readily influenced."?
> >
> > Perhaps this sentence was not well formulated, the intent which is being communicated here is that kinds of data mixed with oasst1 for training had a noticeable effect on the results, which could be seen most prominently in OAIE and VEL evaluation results (Table 1) between the two variants /falcon-40b-sft-top1-560 and /falcon-40b-sft-mix-1226.
> >
> > As for the second part of the question: “What would be the recipe for an intentional change of the nature of the model?”, This is still an open question, but from our (subjective) experience, we recommend first collecting fine-tuning data that exhibits the intended change. Based on this, two approaches exist: Either first fine-tune on the task-specific data, then additionally fine-tune on OASST1, or mix the two datasets and run combined fine-tuning. We have found both methods to work, with none of them emerging as the clear winner for all tasks and datasets.
> >
> >
> >
> > > what are the main limitations and opportunities for improvement for toxicity detection
> >
> > We believe that the correlation data we show indicates that automated and human toxicity detection are aligned to a degree, and therefore, better automated toxicity detection, for example when applied as a tool to assist in content moderation, could be in scope for NLP systems. On the other hand, the absolute numbers of detections are still quite low, which indicates that at the current time, there is still a large volume of data that is classified as inappropriate by humans, but not detected by the automated classifiers. We hope that our dataset can also serve as a basis for the training and/or evaluation of future improved toxicity detectors.
> >
> >
> > > is there a process in place to allow external users and open source community to take part in data filtering so that it does indeed converge to the highest standard?
> >
> > Good question. We already provide part of this process in the data collection mechanism itself, by collecting ratings, up-/down-votes, etc. which already serves as a highly distributed quality annotation process.
> > The specific sentence quoted is to indicate that by releasing the data in a minimally filtered way (we try to only perform filtering deemed as absolutely necessary), we enable every downstream developer to perform their own filtering, and ultimately the open-source community to develop a (distributed) consensus on methods of filtering. We believe outsourcing such highly non-trivial work to the collective will lead to much greater value than if we were to make the decisions for everyone.

---

> > > ### Comment · Reviewer_Tg3T · 2023-08-22
> > > **Response to Authors**
> > >
> > > Dear Authors,
> > >
> > > Thank you for your detailed response, additional results, and provided clarifications. I believe both the paper, the released artifacts, and the rebuttal responses are very valuable to the community and therefore I raise my score.

---

### Author Response · Authors · 2023-08-29
**Final revision with the last few updates**

Dear Reviewers

Together with our previous responses, we've uploaded a revised version of the paper that contained the changes we've referenced in our responses. In those responses, we've promised a few extra improvements. We have now uploaded a new revision containing these extra improvements, plus a few nits. We've added more related work references to the introductory section, specifically more references to work aligning language models to human-collected data, and we've also provided references for the notions of alignment and the prevalence of ChatGPT, which were missing in our original submission. We've also added a paragraph to the appendix giving a brief description of the used evaluation benchmarks.

Thank you all for your feedback, this was very fruitful and we think it served to greatly improve the paper.

---

### Decision · Program_Chairs · 2023-09-22

**Decision:**

Accept (Oral)

**Comment:**

This paper includes a large dataset of dialogue conversations and human feedback.  Key strengths of the paper are:

The dataset was constructed with great care and attention to quality.  Multiple reviewers commented on this: "construction is discussed in great detail, with excellent explanations of all design choices, of the system and UI for annotations, and of the instructions provided for the annotators" and "Rigorous data quality measures, including spam filtering and content moderation on multiple dimensions."

The utility of the dataset is demonstrated via training models with the data and open-sourcing the models, demonstrating that the data can improve performance which reviewers believe will be valuable to the community: "All the code is released and both the training code as well as the provided checkpoints are very valuable for the community."

The dataset is diverse and moves towards further democratizing AI: "scale and diversity of the dataset is impressive" and "unique open crowd-sourced dataset with ensured diversity and transparancy."

Personally, I think this dataset will foster research both in training better models but also understanding human preference data and setting an example of collecting high quality diverse data.

There were some weaknesses pointed out by reviewers, including flushing out the related work section, unclear language, and missing details on things like evaluation metrics.  Authors worked to remedy concerns, and on the whole, I think this paper will make an excellent contribution to the Benchmarks track.

Given the high scores and potential usefulness of this dataset, I recommend it as an oral.